# Alternative Splicing, RNA Editing, and the Current Limits of Next Generation Sequencing

**DOI:** 10.3390/genes14071386

**Published:** 2023-06-30

**Authors:** Manuela Piazzi, Alberto Bavelloni, Sara Salucci, Irene Faenza, William L. Blalock

**Affiliations:** 1“Luigi Luca Cavalli-Sforza” Istituto di Genetica Molecolare, Consiglio Nazionale delle Ricerche (IGM-CNR), 40136 Bologna, Italy; manuela.piazzi@cnr.it; 2IRCCS, Istituto Ortopedico Rizzoli, 40136 Bologna, Italy; 3Laboratorio di Oncologia Sperimentale, IRCCS, Istituto Ortopedico Rizzoli, 40136 Bologna, Italy; alberto.bavelloni@ior.it; 4Dipartimento di Scienze Biomediche e Neuromotorie (DIBINEM), Università di Bologna, 40126 Bologna, Italy; sara.salucci@unibo.it (S.S.); irene.faenza2@unibo.it (I.F.)

**Keywords:** splicing factors, adenosine deamination, cytidine deamination, RNA editing, spliceosome, WGS/WES, RNA-seq, cancer

## Abstract

The advent of next generation sequencing (NGS) has fostered a shift in basic analytic strategies of a gene expression analysis in diverse pathologies for the purposes of research, pharmacology, and personalized medicine. What was once highly focused research on individual signaling pathways or pathway members has, from the time of gene expression arrays, become a global analysis of gene expression that has aided in identifying novel pathway interactions, the discovery of new therapeutic targets, and the establishment of disease-associated profiles for assessing progression, stratification, or a therapeutic response. But there are significant caveats to this analysis that do not allow for the construction of the full picture. The lack of timely updates to publicly available databases and the “hit and miss” deposition of scientific data to these databases relegate a large amount of potentially important data to “garbage”, begging the question, “how much are we really missing?” This brief perspective aims to highlight some of the limitations that RNA binding/modifying proteins and RNA processing impose on our current usage of NGS technologies as relating to cancer and how not fully appreciating the limitations of current NGS technology may negatively affect therapeutic strategies in the long run.

## 1. Introduction

The advent of high-throughput technologies (HTTs) like DNA microarray technology and reverse-phase protein arrays (RPPAs), in the 1990s, revolutionized the manner in which a gene expression analysis was undertaken in the life sciences [1,2]. Prior to HTTs, most biological studies focused on a single gene/protein or a handful of genes with a quantitative reverse-transcription polymerase chain reaction (qRT-PCR) and Northern and Western blotting techniques used to assess gene/protein expression. Over the years, HTTs in nucleic acid analyses further developed to include whole genome sequencing (WGS), whole exome sequencing (WES), whole transcriptome sequencing (WTS or RNA-seq), non-coding RNA-seq, and variations of these techniques to identify modified (acetylated or methylated) nucleic acids [3]. In contrast, HTTs in protein expression analyses developed along two fronts: those involving modifications of RPPA and advancements in mass spectrometry [4,5,6,7]. Modern gene expression analyses now add single-cell and spatial parameters to these HTTs [8,9,10]. But similar to limitations that post-translational modifications (PTMs) impose on a protein expression analysis, data derived from HTTs involving gene expression analyses at the nucleic acid level are limited by a number of factors, some of the most important being the presence of RNA binding proteins (RBPs), the isoforms of these RBPs present, the PTM status of these RBPs, and the activities these RBPs catalyze during RNA processing. Barring epigenetic modifications in DNA, single-cell WGS is currently the most accurate HTT platform. The genome represents the base starting material for gene expression with a limited chromosome/gene copy number, and a single-cell analysis avoids heterogeneity that is often present with an analysis of whole diseased tissue. Still possible though are chromosomal alterations due to duplications, deletions, and translocations that can alter the gene structure and copy number or create novel fusions that cause generated data to deviate from the expected. Added to this is the possible influence of DNA editing enzymes, mutations in genes affecting the efficiency of DNA repair, or innate immune/stress-mediated changes [11,12,13,14]. But, all in all, the genome of a cell is that particular cell’s genome. For the transcriptome, it is not that simple, and it is becoming more evident that expressed transcripts and isoforms of proteins can vary depending on the tissue/cell type, state of differentiation, microenvironment, stress, and disease, with many RNA transcripts and potential protein isoforms going undocumented and unvalidated.

## 2. RNA Modifications

All RNA transcripts can undergo some form of post-transcriptional processing. In eukaryotes, the classic example involves mRNA capping, polyadenylation, and splicing. These processes involve the addition of a N7-methylated guanosine linked 5′-5′ to a second guanosine by a triphosphate linkage (7mGpppG) to the 5′ nucleotide of the pre-mRNA, the addition of a poly A tail to the 3′ end of the sequence, and the removal of introns to form the mature mRNA. Both 7mGpppG capping and polyadenylation influence the stability of the mRNA transcript, its nuclear export and subsequent translation, thus influencing the expression level of the encoded protein [15]. In addition to these processes, a number of additional modifications can be added to all RNA species, including methylations (A, C, and ribose ring), hydroxymethylations (C), and isomerizations of uridine (U) to pseudouridine (Ψ). These modifications can influence the folding and stability of the RNA, the proteins that associate, and the trafficking of the RNA [16,17]. While the exact influence of all these RNA modifications is not completely understood, some such as 2′O-methylation of the ribose ring in the second nucleotide of the mRNA cap appear to serve a purpose in distinguishing self-transcripts from non-self [18,19]. But the fact that these modifications do not directly change the functional transcript sequence limits the influence they impose on an HTT gene expression analysis, and they will not be further discussed here. For more information on these types of modifications, see reviews by Boo and Kim, 2000 and Roundtree et al., 2017 [16,17]. In contrast, the processes of splicing and RNA editing alter the actual sequence, structure, and coding of the RNAs affected. As with other post-transcriptional modifications, a specialized group of RNA binding proteins catalyzes these processes.

## 3. mRNA Splicing and Alternative Splicing

In eukaryotic cells, the removal of introns and the splicing together of exons is mandatory for the proper synthesis of mature mRNA transcripts and the synthesis of encoded proteins. In the nucleus, the association of the spliceosome complex initiates the first post-transcriptional modification of the pre-mRNA. The assembly of the spliceosome complex on the pre-mRNA and subsequent splicing require both *cis*- and *trans*-acting factors. The *cis*-acting factors consist of RNA sequences 5′ and 3′ of the intron–exon junction and the branch-point (BP) region located between 18-40 nucleotides upstream of the 3′ splice site. In addition to these, sequences present in the introns and exons serve as splicing enhancers or silencers based on the RNA binding proteins that preferentially associate at these sequences. The *trans*-acting factors consist of the small nuclear ribonucleoproteins (snRNPs) U1, U2, U4/6, and U5 as well as close to 300 additional proteins [20,21].

Splicing initiates with the ATP-dependent binding of U1 snRNP to the 5′ splice site of the intron. This interaction is stabilized by members of the serine/arginine-rich (SR) protein family. Following this initial step, SF1/BBP binds the BP region while the 35-kDa subunit of the heterodimeric U2 auxiliary factor (U2AF) binds the 3’ splice site. The 65-kDa subunit of U2AF then serves to bridge SF1/BBP and the 35-kDa subunit of U2AF. This initial recognition of 5′ and 3′ splice sites forms the E-complex. At this juncture, the U2 snRNP associates in an ATP-dependent manner with the BP forming the pre-spliceosome (A-complex) and is stabilized by the association of SF3 (A and B) proteins as well as U2AF. SF1/BBP is then displaced by other SF3 proteins. A pre-assembled U4/U6/U5 complex is then recruited to the A-complex to form the pre-catalytic spliceosome (B-complex). Structural alteration of the spliceosome results in the destabilization of the U1 and U4 interactions, resulting in the loss of these snRNPs and the generation of an active spliceosome complex or B’-complex, which initiates the first catalytic step of splicing to form the C-complex [20,21]. The second catalytic step in splicing occurs through structural modification of the spliceosome RNP component, which may associate or disassociate from the complex. The final phase results in the release of U2, U5, and U6 snRNPs and the release of the spliced mRNA and intron (Figure 1). This process sequentially occurs to form the mature mRNA, but this is not always the case. Statistical data suggest that, on average, a gene encoding 11 exons will produce approximately 5.4 differing mRNAs. The responsible process is referred to as “alternative” splicing [22]. 

Alternative splicing of mRNA is a fundamental process to enhance the possibility of gene expression and increase protein diversity. It is estimated that at least 50% of genes express alternately spliced isoforms that vary according to the tissue/cell type and condition [21]. This estimate may be low as many immune-related genes have been observed to undergo varying degrees of alternative splicing relative to conditions. In T- and B-lymphocytes, RNA-seq and microarray studies suggested that nearly 60% of all genes expressed were alternately spliced [23]. Five patterns of alternative splicing have been described: exon skipping, intron retention, mutually exclusive exons, alternative 5′ splice site, and alternative 3′ splice site (Figure 1). The major players involved consist of the *trans*-regulatory factors, serine/arginine-rich splicing factors (SRSF) and heterogeneous nuclear ribonucleoproteins (hnRNPs), and the *cis*-regulatory elements defined as exonic splicing enhancers (ESE), intronic splicing enhancers (ISE), exonic splicing silencers (ESS), and intronic splicing silencers (ISS). Usually, SRSF proteins associate with ESE and ISE elements, favoring the recruitment of U1 and U2 snRNPs and auxiliary factors and subsequent splicing, while in contrast, association of hnRNPs to ESS and ISS elements favors suppression by inhibiting spliceosome access to the polypyrimidine tract [20,24]. Impairment of alternative splicing is associated with a failure of normal cellular function with a final outcome of disease [20,21]. It is becoming more evident that for a majority of genes, a single primary transcript does not code simply for a single protein but rather diverse isoforms of the same protein, each having specific functions in given tissues. Not only can alternative splicing affect the amino acid sequence of the resulting protein, affecting its function/activity or localization, but it can also influence the translation efficiency and stability of the encoding mRNA and thus regulate protein expression post-transcriptionally but pre-translationally. This mechanism is finely tuned at developmental stages in different tissues, and an alteration in regulation of alternative splicing is now linked with several human diseases, including pre-leukemic states such as MDS, leukemia, a variety of solid tumors, Alzheimer’s disease, Parkinson’s disease, and metabolic and autoimmune diseases [20,21,25]. Identification of frequent mutation of genes involved in RNA splicing in cancer gives an interesting view into the mechanism of cancer-specific alternative splicing [21,25]. As changes in alternative splicing and regulators of alternative splicing have been associated with disease, it is only reasonable to assume that diseased tissues present with a significant percentage of abnormal transcripts, many most likely yet identified or validated, contributing to their absence in curated databases (NCBI, ENSEMBL) and their elimination by most standard expression profiling protocols that map reads to annotated transcripts. But these transcripts may be highly important and have roles (positive or negative) in the disease state.

### 3.1. Serine/Arginine-Rich Splicing Factors (SRSFs)

The serine/arginine-rich (SR) splicing factor family is comprised of 12 phylogenetically conserved and structurally related proteins encoded by genes, *SRSF1-12*, scattered throughout the genome (Figure 2A). These proteins act in complexes to control constitutive and alternative pre-mRNA splicing, as well as in other aspects of gene expression (Figure 2A, Table 1) [26]. SRSFs contain one or two RNA-recognition motifs at the N-terminus and one SR-rich domain, which serves as an intermediary in the interaction with other proteins, at the C-terminus. SRSF1 and SRSF2, which have been extensively investigated, serve pivotal roles in cell cycle regulation, genome stability, and translation as well as pre-mRNA splicing, stability, and transport [26,27]. Most SRSFs are localized to the nucleus, nucleoplasm, or speckles, but a number are also found in the cytoplasm and shuttle between the cytoplasm and nucleus.

The activity of the SRSF proteins is highly regulated by extensive and reversible phosphorylation of serine residues carried out by several kinases, including AKT family kinases, CDC-like kinases (CLKs), SRSF protein kinases (SRPKs), pre-mRNA splicing 4 kinase (PRP4K), and protein kinase A (PKA) [28,29,30]. These phosphorylations modulate protein–protein interactions within the spliceosome and regulate the activity and sub-cellular distribution of SRSF proteins. Therefore, changes in the phosphorylation state of SRSFs during growth factor/cytokine stimulation or stress (inflammation, cytotoxic cytokines) play a critical role in the control of their activity and the splicing landscape of expressed transcripts. Other than phosphorylations, additional PTMs, such as acetylations, methylations, succinylations, sumoylations, and ubiquitinations, are present. But with the exceptions of methylations at the amino-terminus, which appear to regulate subcellular localization to the nucleus, and ubiquitinations at the amino-terminus, which appear to promote protein degradation, the significance of most of these PTMs is unknown (Table 1) [31]. SRSFs are also known to influence splicing of their own transcripts and that of other SRSFs, thus resulting in multiple SRSF isoforms that most likely demonstrate altered activity and transcript/substrate preference; the majority of these alternative isoforms of SRSF proteins have never been examined. Moreover, the activity of these SRSFs is also altered by other splicing modulators such as splicing regulatory glutamine/lysine-rich protein (SREK)-1, serine/arginine repetitive matrix protein (SRRM)-1/-2, cytotoxic granule-associated RNA binding protein TIA1, and transformer-2 protein homolog (TRA2)-A/B (Table 1) [32,33,34,35,36]. Thus, differential expression of these splicing factors, the alternate isoforms expressed, and their PTMs, mutations, and regulation by additional splicing factors each can drastically change the RNA splicing landscape to produce an infinite number of potential transcripts and protein isoforms that have specific roles. 

### 3.2. Heterogeneous Nuclear Ribonucleoproteins (hnRNPs)

Heterogeneous nuclear ribonucleoproteins assist in the maturation of pre-mRNA transcripts to mature mRNAs during nuclear to cytoplasmic transport, and their subsequent translation, by influencing alternative splicing and mRNA stability, transport, and folding. There are around 20 major types of hnRNPs and several minor families whose genes are scattered throughout the genome, some being sex-linked (Figure 2B, Table 2) [37]. The hnRNPs contain a nuclear localization sequence (NLS) and are thus primarily localized to the nucleus; but this localization is highly dependent on post-translational modifications, which regulate the nuclear–cytoplasmic localization as well as intermolecular interactions of the hnRNPs [37]. In addition to the NLS, the hnRNPs contain at least one or a combination of four types of RNA-binding domains (RBDs): RNA recognition motif (RRM), RRM-like, glycine-rich with an RGG box, or KH domain. The specificity of RNA–protein binding within hnRNPs is entirely dependent on the three-dimensional structure of the protein surrounding the RBD, with diversity dictated by the combination of different RNA-binding motifs present [37].

Similar to SRSF proteins, a number of hnRNPs also express alternatively spliced isoforms, which potentially change activity or substrate specificity, and PTMs, including acetylations, caspase cleavage (activation/protein processing), methylations, N-glycosylations, *O*-linked β-*N*-acetylglucosaminylation (O-GlcNAc), phosphorylations, succinylations, sumoylations, and ubiquitinations, whose roles in the cell are not clear (Table 2) [31]. Many of the hnRNPs are known to be part of the spliceosome complex and influence splicing of particular mRNAs or their alternative splicing; thus, alternations in their expression, isoform expressed, PTMs, and with what proteins they interact can change the expressed RNA landscape. In addition to splicing and alternative splicing, hnRNPs are known to carry out a number of other functions: (1) binding of certain hnRNPs to the 3’-end untranslated region (UTR) stabilizes some mRNAs while it targets the degradation of others; (2) certain hnRNPs sequester specific mRNAs or suppress their translation; (3) hnRNPs can influence internal ribosome entry site (IRES)-dependent translation of specific mRNAs; and (4) hnRNPs can also influence miRNAs [37]. The expression of hnRNPs is known to influence the transcription and translation of multiple oncogenes and tumor suppressors, epithelial–mesenchymal transition (EMT), and immune/inflammatory regulation; thus, it is not surprising that alterations/mutations in a number of these proteins are associated with disease [37]. As with the alternative SRSF isoforms, the cellular functions of most alternative hnRNP isoforms have never been examined, nor is it clear what the influence of PTMs to the hnRNPs is on alternative splicing. Thus, the potential exists for hnRNPs, either through differential expression, isoforms expressed, or PTM regulation or mutation, to produce an infinite number of potential transcripts and protein isoforms that have specific roles but have not yet been identified or annotated.

### 3.3. Consequences of Alternative Splicing and Measures to Address the Issue

The biggest issue involving alternative splicing in regards to the interpretation of RNA-seq data is the shear absence of many alternatively spliced transcripts from the databanks. This stems from both the inability to continuously update the databanks, as well as the failure of the research community to timely upload novel transcripts when found, which of course may occur for any number of understandable reasons. As can be seen in the previous sections, the number of protein complex combinations, isoforms involved, and the status of their PTM suggest, in theory, an infinite number of splicing possibilities across the transcriptome. Certainly, many of these may not lead to viable mRNA products as they are relegated to nonsense mediated decay or encode a protein product that is highly unstable and rapidly degraded, but it would be premature to suggest that these products do not have a function in the cell. Twenty years ago, pseudogenes were viewed as leftovers/byproducts of gene amplification; only later was it found that in many cases, these non-coding mRNAs acted as bait to sequester miRNAs away from the actual coding transcripts (primary examples include the tumor suppressor *PTEN*). One must also keep in mind that as well as having an infinite number of splicing possibilities, we also have an infinite possibility of conditions that may influence alternate splicing and promote the production of novel isoforms. Moreover, the response to these conditions and the expression of alternate-spliced isoforms may be, and likely is, tissue/cell-type specific and highly influenced by the disease state of the tissue. Thus, it is not surprising that alterations in several splicing factors (SRSFs and hnRNPs; Table 1 and Table 2) are associated with disease and the expression of these factors associated with cancer. Therefore, while the physical long-read RNA sequencing itself may catch all these isoforms, a standard analysis protocol is bound to miss the majority of novel variants. So how can the presence of undocumented splice variants be verified?

For starters, little information can be gained from the gene expression profile data itself (gene/isoform quantification), and while many gene profiles from gene expression arrays and RNA-seq are available as open access, the information they provide in this context is only partial. One of the more direct methods is manually assembling transcript products from a specific gene. Manual mapping is a low-throughput method but has the highest probability of identifying all transcript products from an individual gene. In addition, validation and cloning of any alternate spliced forms identified can be undertaken with RNA isolation, gene-specific amplification of transcripts, and RNA-seq together with shotgun cloning and DNA-seq, giving a representation of all transcripts present that are related to a particular gene. Of course, this works for when the expression of only a handful of genes is of interest [38]. When it comes to identifying novel splice variants from RNA-seq on a larger scale, the past several years have seen a vast increase in the number of algorithms to detect novel alternately spliced transcripts. In most cases, these algorithms bypass the direct use of annotated transcript databases such as ENSEMBL or NCBI’s RefSeq for transcript identification but may still use these annotated sequences as a scaffold for assembling reads or just comparison [39,40]. 

## 4. RNA Editing and RNA Editing Enzymes

RNA editing is an important post-transcriptional mechanism occurring in a wide range of organisms, which alters the primary RNA sequence through the insertion/deletion or modification of specific nucleotides. The most common modification in humans is deamination, leading to both a biochemical and functional change in the nucleotide [41]. In RNA, two forms of nucleotide deamination are observed and involve either adenosine or cytidine residues. Deamination of adenosine at the C6 position results in the biochemical conversion of adenosine to inosine. As inosine is interpreted by the cellular machinery as guanine, deamination also results in the functional equivalent of an adenosine to guanine conversion in the affected transcript. Similarly, deamination of cytidine at the C4 position results in the biochemical and functional conversion of cytidine to uridine (Figure 3).

Deamination of adenosine and cytidine residues in RNA is carried out by dedicated and specific enzymes classified according to the activity they catalyze as either adenosine deaminases or cytidine deaminases. The adenosine deaminases consist of the adenosine deaminase acting on the double-strand RNA (ADAR) family and the adenosine deaminase acting on the tRNA (ADAT) family, while the cytidine deaminases belong to the activation-induced cytidine deaminase/apolipoprotein B mRNA editing enzyme and the catalytic polypeptide (AID/APOBEC) family (Figure 3, Table 3).

Beyond modifying an RNA secondary structure, protein binding and changing protein coding and adenosine deamination are also known to alter splice donor–acceptor site selection by modifying key adenosine residues near the acceptor site, thus promoting alternative splicing of transcripts. In non-coding RNA, such as miRNAs, deamination can result in alteration of the seed region, thereby changing the target specificity of miRNAs. Similarly, cytidine deamination alters transcript coding, structure, and stability and may alter miRNAs, although generally at a much lower frequency compared to adenosine deamination. Therefore, the enzymes that carry out these activities have the potential to drastically enhance transcript variability on a small or large scale, depending on the tissue, process involved, and conditions, as well as promote disease (Figure 1) [42,43]. Moreover, the enzymes that principally carry out these modifications are also associated with the genomic stability, DNA repair, and modification of nucleotides in genome-associated ssRNA/ssDNA, which potentially alter the genome [11,12,44,45,46,47].

### 4.1. Adenosine Deaminases

The majority of RNA editing in mammalian cells is carried out by the ADAR family of enzymes. This family consists of three independently encoded proteins, ADAR1 (*ADAR*; 1q21.3), ADAR2 (*ADARB1*; 22q22.3), and ADAR3 (*ADARB2*; 10q15.3). These enzymes are composed of a series of tandemly arranged double-strand (ds) RNA binding domains (dsRBDs; 3 in ADAR1 and 2 in both ADAR2 and ADAR3) upstream of the catalytic domain. In addition, full-length ADAR1 also contains two Z-α domains in the amino terminus, which allow ADAR1 to associate with nucleic acids possessing a left-handed helical structure (Z-RNA/Z-DNA) (Figure 3A) [48]. ADAR1 and ADAR2 are ubiquitously expressed, functional deaminases, while ADAR3 is catalytically inactive with expression mostly restricted to the brain and neural tissue [42,49]. These enzymes act on RNAs as either homo- or heterodimers, and the diverse dimeric combinations are believed to influence substrate selection and modification. In addition, each of these genes produces a number of protein coding isoforms through alternative splicing (ADAR1 (5 isoforms), ADAR2 (6 isoforms), and ADAR3 (2 isoforms)), or through the use of alternative transcriptional start elements that alter exon 1 (ADAR1 (2 isoforms)) [50,51,52]. Of these, only a handful have been fully characterized. The two prominent forms of ADAR1 expressed in the cell, ADAR1p110 and ADAR1p150, are produced through the use of alternative promoter/transcriptional start sites and alternative splicing. Transcriptional initiation at promoters 2 and 3 result in the inclusion of an exon 1 (1B and 1C) devoid of an AUG start codon so that translation of the final mRNA product initiates in exon 2, thus producing ADAR1p110. In contrast, transcriptional initiation at promoter 1, which is interferon-responsive, results in the inclusion of an exon 1 that contains an AUG start codon, thus encoding a protein containing an additional 295 amino acids, ADAR1p150 [52,53]. As ADAR1p110 primarily localizes to the nucleus and ADAR1p150 primarily localizes to the cytoplasm, the RNA substrates that these isoforms preferentially modify likely differ. Moreover, of the reported isoforms of ADAR1 generated by alternative splicing, many demonstrate spatial changes in the structural distribution between dsRBDs or between dsRBD 3 and the catalytic domain, suggesting that these isoforms may have different substrates than those thus far identified for ADAR1p150 and p110, further enhancing genetic variability.

Similar to ADAR1, two main forms of ADAR2 have been characterized, ADAR2 long (ADAR2) and ADAR2 short (ADAR2-S). ADAR2-S results from the alternative splicing of an Alu element, producing an ADAR2 isoform that is missing amino acids 466-505. The lack of these 40 amino acids in ADAR2-S results in an enzyme with elevated catalytic activity. In contrast, only one form of ADAR3 has been significantly characterized [50,51,54,55,56,57].

Beyond alternative splicing and homo-/heterodimerization, these editases are regulated by a number of post-translational modifications. In ADAR1p150, over 120 PTMs have been identified, mostly through high-throughput methods like mass spectrometry, but only a few of these PTMs have been characterized for their effect on ADAR1 activity [31]. Sumoylation of K418 inhibits editase activity while ubiquitination of K574 and K576 promotes proteosome-mediated degradation of ADAR1 [58,59]. Phosphorylation of T808, T811, S823, and S825 through stress activation of the MKK6-p38-MSK MAP kinase pathway promotes the nuclear export of ADAR1p110, while phosphorylation of T1033 alters editase activity toward specific substrates [60,61]. Phosphorylation of T1033 is rather interesting as it suggests that PTMs may not be all or nothing (activating editase activity or inhibiting editase activity) but may alter editase activity toward specific targets, changing the repertoire of edited transcripts [42,61]. For ADAR2, 23 PTMs have been identified, again mostly with high-throughput methods [31]. Like ADAR1, the effect that most of these PTMs have on ADAR2 is unknown. To date, only three of these PTMs have been characterized. Phosphorylation of S211 and S216 in the spacer region between dsRBD2 and the editase domain by PKCζ results in enhanced editing activity, while similar to phosphorylation of T1033 in ADAR1, phosphorylation of the homologous site in ADAR2, T553, alters editase activity toward specific substrates [61,62]. Hence, other than alternate isoforms of these enzymes, which likely display a differing substrate preference and activity, regulation by PTMs can also influence the enzymatic complexes with which these editase participate, substrate preference, and activity; thus changing the panorama of transcripts altered.

Unlike the ADAR family, ADAT proteins, ADAT1, ADAT2, and ADAT3, carry out a minor portion of the RNA editing in the cell. As their name indicates, editing catalyzed by these enzymes is specific to tRNA, thus their influence is rather specific to translation and has no significant impact on a DNA/RNA-seq analysis and will not be further discussed here [63].

### 4.2. Cytidine Deaminases

In humans, cytidine deamination of RNAs is carried out by certain members of the APOBEC family. The family consists of APOBEC1, APOBEC2, APOBEC3 (3A, 3B, 3C, 3D, 3F, 3G, 3H), APOBEC4, and activation-induced cytidine deaminase (AID). Similar to ADAR family proteins, the subcellular localization of APOBEC proteins dictates target selection. APOBECs 1, 2, 3A, 3C, 3G, and 3H are each cytosolic and nuclear with an enrichment of 3G and 3H in processing bodies (P-bodies). APOBEC3B is strictly nuclear while APOBECs 3D and 3F are cytosolic and often enriched in P-bodies [12,64]. The sub-cellular localization of APOBEC4 has not been determined, but prediction in silico suggests a predominant nuclear localization. To date, with the exception of AID and APOBEC-3B, -3F, and -3G, little information exists on the post-translational modification/regulation of these proteins [31]. PKA-dependent phosphorylation of amino-terminal amino acids in AID (T27 and T38) and APOBEC3G (T32) was reported to result in enzymatic activation [65,66]. In contrast, PKA-dependent phosphorylation in the NAD2 domain of APOBEC-3B (T214), -3F (S216), and -3G (T218) was reported to inhibit activity [66,67,68].

Although the first members of this family were identified for their RNA editing ability, not all members of this family edit RNA. APOBEC proteins function as monomers, homodimers, heterodimers, and tetramers depending on the family member (Table 3) [11]. The first family member identified, APOBEC1 (12p13.31), was discovered for its ability to catalyze the cytidine deamination of apolipoprotein B (APOB) mRNA and generate a stop codon at amino acid 2180, thus generating a truncated form of APOB, APOB48 [69]. In normal human tissues, APOBEC1 expression is limited to the small intestine where it is known to only edit APOB, while in mice, numerous APOBEC1-edited mRNAs have been identified along with a variety of cofactors that dictate mRNA substrate specificity. A significant amount of data from mice have suggested the importance of APOBEC1-dependent editing in immunity [11,12,70,71]. Interestingly, APOBEC1 expression and editing is enhanced in several human cancers, but this association appears to be more related to the ability of APOBEC1 to deaminate DNA rather than RNA [11,12,72,73,74].

Mostly expressed in lymphoid tissues and B-lymphocytes, AID (12p13.31) is a ssDNA deaminase that is responsible for Ig heavy chain class switching and hypermutations in variable regions, promoting antibody maturation [75]. Localization of AID is predominantly cytosolic with nuclear localization dependent on germinal center-associated nuclear protein (GANP) and β catenin-like protein 1 (CTNNBL1) [76,77]. Like many other APOBEC family members, off-target editing of the genome by AID has been reported [78,79]. Although AID binds both single-stranded (ss) DNA and ssRNA, it is only catalytically active toward ssDNA.

In mice, APOBEC3 is encoded in a single gene that diverges and expands in humans to seven related genes in a cluster located on chromosome 22 (22q13.1). APOBEC3 proteins are widely expressed in tissues. Expression is most noted in lymphoid tissue and peripheral blood mononucleated cells as well as reproductive organs, suggestive of a significant role in innate immune signaling. While certain APOBEC3 members (3A and 3G) can edit ssRNAs, the primary purpose of these editases is the modification of ssDNAs arising during viral infection [11,12]. In addition to editing viral ssRNAs (3A and 3B) and ssDNAs (3A, 3B, 3C, 3D, 3F, 3G, and 3H), APOBEC3 members have also been observed to modify host cell ssRNAs (3A and 3G) and ss genomic DNAs (3A and 3B) at sites of DNA damage repair [11,80,81,82,83,84]. Enhanced activity of these enzymes has been reported to promote DNA damage and genomic instability.

In contrast to other APOBECs, the expression of APOBEC2 (6p21.1) is restricted to cardiac and skeletal muscle. APOBEC2 does not demonstrate any deaminase activity toward DNA or RNA yet strongly binds to DNA at specific promoters and is suspected of being a transcriptional repressor that regulates myoblast differentiation and myogenic stem satellite cell self-renewal [85,86]. Finally, the expression of APOBEC4 (1q25.3) is restricted to the testis and appears to have a role in epigenetic remodeling of promoter regions [87,88]. An interesting aside to several of these cytidine deaminases (3A, 3B, 3C, 3F, and 3G) as well as ADAR1p150 is their ability to be induced by an interferon; thus, activation of the innate immune system and the integrated stress response (ISR) promotes their expression, thereby enhancing editing of both transcripts and potentially genomic DNA. Similar to the ADAR proteins, a number of splice variants have been identified (and likely many others unidentified) for various AID/APOBEC family members (Table 3). Like with ADAR proteins, these alternately spliced forms are expected to demonstrate altered activity/substrate specificity in comparison to the canonical form, adding additional complexity to how these RNA editases may diversify the transcriptome and/or alter the transcriptome to genome homology.

### 4.3. Consequences of RNA Editing and Measures to Address the Issue

RNA editing presents many of the same issues that alternative splicing presents but with the potential for modifications much more difficult to clearly identify. Both WGS and single-cell techniques can remove a significant amount of potential variability in regards to editing. Our current understanding of the ADAR family is that adenosine deaminations catalyzed by ADAR1 and ADAR2 are restricted to RNA, although the association of ADAR1p110 and ADAR2 with DNA repair complexes and their involvement in the DNA repair process should at least be mentioned for consideration [44,45,46,47]. In contrast, APOBEC family members target genomic DNA and RNA; thus, transcript changes visualized in a complex multicellular analysis may result from genomic or transcriptomic changes and would require WGS to rule-out any germ line changes. A single-cell analysis would reduce this variability and allow the origin of changes (genome or mRNA) to be determined based on the number of reads containing a particular edit. Single nucleotide changes induced by RNA editing can influence alternative splicing but can also cause punctiform modifications/mutations to the encoded protein. While it is difficult for splicing/alternative splicing to be an artifact, the case is different regarding point mutations. Again, the frequency of edits becomes highly important as artifacts in the nucleotide sequence are common. As with SNPs, the frequency of punctiform changes must be determined to distinguish an artifact from reality. Most pipelines include read alignment, and thus, individual counts of edits are normally analyzed or can be easily analyzed from the raw data. While methods to determine the identity of novel splice variants resulting from RNA editing are no different than those discussed above regarding splicing factors, specifically determining a role for RNA editing in the alternative splicing of particular transcripts is not a direct process. Sites of deamination influencing alternative splicing are often located in introns and are lost from the final product during the splicing event and thus difficult to define unless caught in edited pre-mRNA. In contrast, those affecting acceptor/donor sites are usually quickly identifiable from the sequence but still serve a level of manual surveillance to catch. To date, the best modes of defining alternative spliced transcripts related to RNA editing are first identifying the novel transcript and any noted sequencing differences, and then individually analyzing the presence of known RNA editing sites in the gene of suspect novel transcripts, using the A-to-I REDIportal database [89]. Currently a comprehensive database for C-to-U editing does not exist. Moreover, when a high percentage to a complete switch of a particular splice variant is observed, DNA editing by APOBEC family members should also be considered, as changes in the genomic sequence also have the potential to influence downstream splicing with particular alternative events occurring at a much higher frequency.

## 5. Conclusions

Increasing use of high-throughput technologies for diagnoses, prognoses, and therapeutic responses of patients often has overlooked caveats…the transcriptome does not necessarily reflect the genome, nor will all transcript reads be correctly identified. Early studies using genome arrays based on short probe binding missed many alternatively spliced transcripts, and of the transcripts they recognize, the assay format was not adequate to distinguish the presence or absence of most alternatively spliced transcript isoforms, nor were these formats adequate at catching post-transcriptional edits of the transcripts. Using this approach, multiple transcript isoforms may bind to the same oligonucleotide probe and vice versa; some expressing alternatively spliced transcripts may exclude the sequences necessary to bind to a gene’s specific oligonucleotide probe, and thus, their expression fails to be identified. The re-use of most older genome array datasets requires updating/re-aligning the oligonucleotide probes to the genome prior to re-analyzing as well as a focused analysis of oligonucleotide probes demonstrating no significant homology to the genome but demonstrating significant change during the assay.

With the advancement of RNA-seq technology that provides longer reads, many of these points are now resolvable but require significant attention on the part of those conducting the analysis. A streamlined RNA-seq analysis often uses database matching with annotated transcripts reported by the NCBI or ENSEMBL, but unique transcripts in an analysis that have yet to be annotated are not assigned and “discarded”. Alternative splicing resulting from the altered expression/regulation of splicing-associated proteins or RNA editing enzymes can have a significant effect on this aspect. Since many splicing factors and RNA/DNA editing enzymes are affected by and regulate the innate immune/inflammatory response, the effects elicited by alternative splicing factors and RNA editing enzymes become even more applicable under inflammatory/cell stress conditions where innate immune signaling and activation of the integrated stress response become paramount. In addition, high-throughput techniques are not designed to analyze the potential downstream effects that alternative splicing or single-nucleotide editing of the RNA have on the function of specific proteins. Single nucleotide changes observed in RNA-seq need to be defined and compared to WGS/WES to differentiate RNA or DNA editing from polymorphisms (both ADAR and AID/APOBEC family members can promote DNA modifications, which may have a role in adaption/evolution as well as disease) [14]. Thus, in most cases, singular datasets represent a partial analysis painting less than half the picture. Current advancements in RNA-seq bioinformatics will hopefully fill this chasm of information by having the ability to run predicted homologies to assign transcripts currently relegated to experimental “garbage” into meaningful data [39,40]. The next big step will be linking the analysis of the reads with the predicted protein structure and function and the potential impacts of edited sites on protein function and/or targeting of miRNA, finally giving a much clearer picture of what is happening in the cell.

## Figures and Tables

**Figure 1 genes-14-01386-f001:**
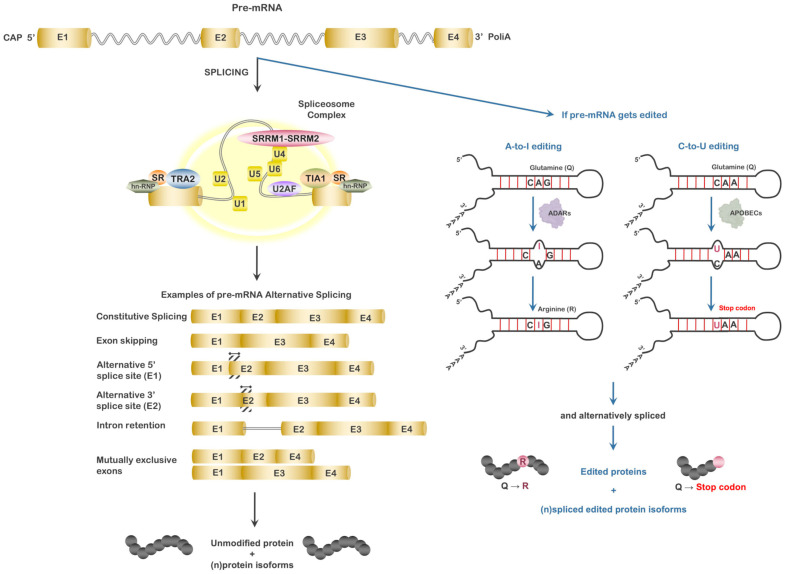
The influence of mRNA splicing and alternative splicing on gene expression. Left panel: Schematic of pre-mRNA splicing and the different forms of alternative splicing. Splicing of pre-mRNA can result in the expression of the full encoded protein or (n) number of protein isoforms produced as a result of alternative splicing. Right panel: Schematic of RNA editing’s potential to alter gene expression. Editing of the pre-mRNA by adenosine or cytidine deaminases can result in single amino acid substitutions to the protein, alter splice site selection (promoting alternatively spliced mRNAs and protein isoforms), or both.

**Figure 2 genes-14-01386-f002:**
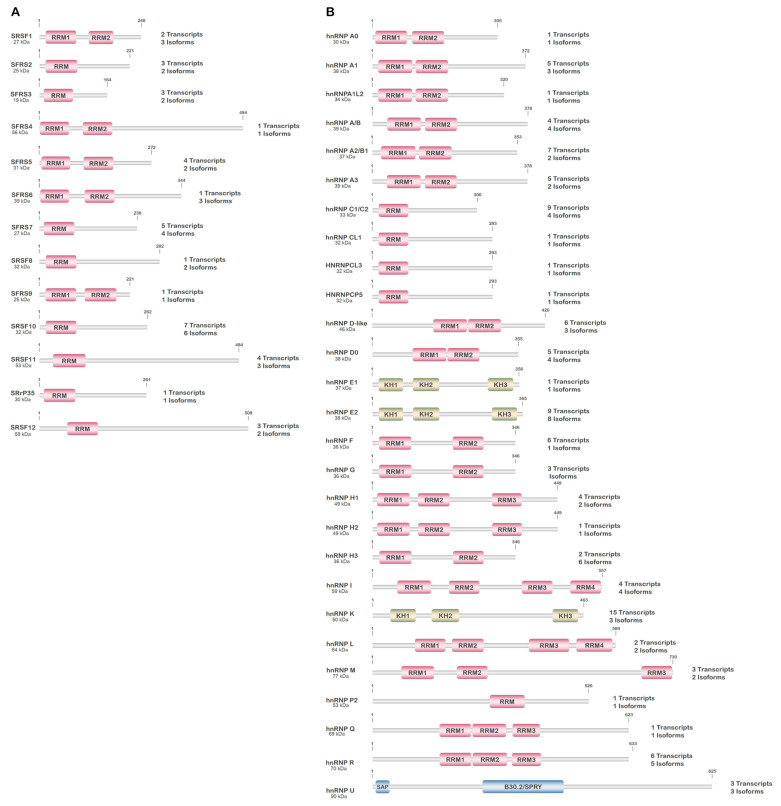
Schematic diagram of the main SRSF (**A**) and hnRNP (**B**) proteins involved in splicing/alternative splicing. The molecular weight (designated main isoform), major structural domains, and the number of validated mRNA transcripts/protein isoforms produced from the respective genes are presented.

**Figure 3 genes-14-01386-f003:**
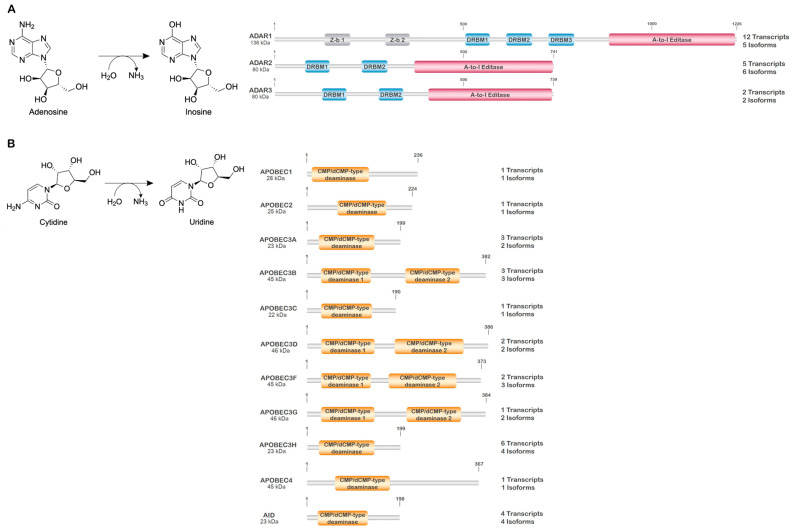
Schematic diagram of ADAR and AID/APOBEC family proteins. (**A**) The deamination of adenosine to inosine is carried out by homo- and heterodimers of the ADAR family proteins. Two main isoforms of ADAR1 (p110 and p150) and ADAR2 (ADAR2long and ADAR2short) are expressed to some level in all cells; only ADAR1p150 and ADAR2short are presented here. ADAR3 is catalytically inactive. (**B**) The deamination of cytidine to uridine is carried out by members of the AID/APOBEC family, functioning as monomers, homodimers, heterodimers, and tetramers depending on the family member involved. The molecular weight (designated main isoform), major structural domains, and the number of validated mRNA transcripts/protein isoforms produced from the respective genes are presented.

**Table 1 genes-14-01386-t001:** Serine/arginine-rich splicing factors: Mediators of alternative splicing.

*Protein* (*UniProt ID*)	*Gene*	*Location*	*Function*/*Role*	*Localization*	^ 1 ^ *Alternant* *Transcripts*/*Isoforms*	^ 2 ^ *PTMs*
*** SREK1** **(Q8WXA9)**	*SREK1*	*Ch.5q12.3*	*Regulates alternative splicing by influencing the activity of other splicing factors. Inhibits the splicing activity of SFRS1, SFRS2, and SFRS6, but augments the splicing activity of SFRS3.*	*Nuc*	***3*** *trscpts/**2** isoforms*	***18*** *(uc)*
*** SRRM1** **(Q8IYB3)**	*SRRM1*	*Ch.1p36.11*	*Part of the pre- and post-splicing mRNP complex. Promotes constitutive ESE-dependent splicing by bridging sequence-specific SRSFs, TRA2B, and basal snRNP factors. Ubiquitously expressed.*	*Nuc matrix, Speckles*	***1*** *trscpt/**2** isoforms #*	***166*** *(uc)*
*** SRRM2** **(Q9UQ35)**	*SRRM2*	*Ch.16p13.3*	*A required component of the pre-mRNA spliceosome. Mainly expressed in liver placenta and leukocytes. Associates with SRRM1, SRSF3, SRSF4, and SRSF5.*	*Nuc, Speckles*	***3*** *trscpts/**3** isoforms*	***736*** *(uc)*
**SRSF1** **(Q07955)**	*SRSF1*	*Ch.17q22*	*Prevents exon skipping to ensure the accuracy of splicing and alternative splicing by forming a bridge between the 5′- and 3′-splice site binding components, U1 snRNP, and U2AF. **Isoform 1** acts as a splicing enhancer. **Isoforms 2 and 3** act as splicing repressors.*	*Speckles, Cyto (shuttles)*	***2*** *trscpts/**3** isoforms*	***65*** *(muc)*
**SRSF2** **(Q01130)**	*SRSF2*	*Ch.17q25.1*	*Central to pre-mRNA splicing. Required for the formation of the ATP-dependent splicing complex. Forms a bridge between the 5′- and 3′-splice site binding components, U1 snRNP, and U2AF.*	*Nuc, Nucleoplasm, Speckles*	***3*** *trscpts/**2** isoforms*	***45*** *(muc)*
**SRSF3** **(P84103)**	*SRSF3*	*Ch.6p21.31-21.2*	*Specifically promotes exon inclusion during alternative splicing. Recruitment of SFRS3 to mRNA binding elements near N6-meythyladenosine (m6A) sites, leads to exon inclusion.*	*Nuc, Cyto, Speckles*	***2*** *trscpts/**2** isoforms**	***33*** *(muc)*
**SRSF4** **(Q08170)**	*SRSF4*	*Ch.1p35.3*	*Plays a role in alternative splice site selection during pre-mRNA splicing.*	*Speckles*	***1*** *trscpt/**1** isoform*	***67*** *(uc)*
**SRSF5** **(Q13243)**	*SRSF5*	*Ch.14q24.1*	*Plays a role in constitutive splicing and can influence alternative splice site selection.*	*Nuc*	***4*** *trscpts/**2** isoforms #*	***39*** *(uc)*
**SRSF6** **(Q13247)**	*SRSF6*	*Ch.20q13.11*	*Plays a role in constitutive splicing and can influence alternative splice site selection.*	*Nuc, Speckles*	***1*** *trscpt/**3** isoforms #*	***56*** *(uc)*
**SRSF7** **(Q16629)**	*SRSF7*	*Ch.2p22.1*	*Required for pre-mRNA splicing. Can modulate alternative splicing in vitro. Prevalent in brain, kidneys, and lungs.*	*Nuc, Cyto*	***5*** *trscpts/**4** isoforms #*	***53*** *(uc)*
**SRSF8** **(Q9BRL6)**	*SRSF8*	*Ch.11q21*	*Involved in pre-mRNA alternative splicing. Highly expressed in pancreas, spleen, and prostate; poorly expressed in lungs, liver, and thymus.*	*Nuc*	***1*** *trscpt/**2** isoforms #*	***44*** *(uc)*
**SRSF9** **(Q13242)**	*SRSF9*	*Ch.12q24.31*	*Plays a role in constitutive splicing and can influence alternative splice site selection. Highly expressed in heart, kidneys, pancreas, and placenta; poorly expressed in brain, liver, lungs, and skeletal muscle.*	*Nuc*	***1*** *trscpt/**1** isoform*	***49*** *(uc)*
**SRS10** **(O75494)**	*SRSF10*	*Ch.1p36.11*	*Acts as a general repressor of pre-mRNA splicing when in its dephosphorylated form by interfering with the U1 snRNP 5′ splice recognition of SNRNP70. Required for splicing repression in M-phase cells and after heat shock. Promotes exon skipping during alternative splicing. May be involved in the regulation of alternative splicing in neurons, with Isoform 1 acting to promote alternative splicing and Isoform 3 acting to suppress alternative splicing.*	*Speckles, Cyto*	***7*** *trscpts/**6** isoforms*	***46*** *(muc)*
**SRS11** **(Q05519)**	*SRSF11*	*Ch.1p31.1*	*May function in pre-mRNA splicing.*	*Nuc*	***4*** *trscpts/**3** isoforms*	***58*** *(uc)*
**SRS12** **(Q8WXF0)**	*SRSF12*	*Ch.6q15*	*Acts as an antagonist to SR proteins in pre-mRNA splicing regulation. Mainly expressed in testis.*	*Nuc*	***1*** *trscpt/**1** isoform*	***21*** *(uc)*
*** TIA1** **(P31483)**	*TIA1*	*Ch.2p13.3*	* Regulates alternative pre-RNA splicing by binding U-rich sequences immediately downstream of 5′ splice sites in a U snRNP-dependent manner. Can promote atypical 5′ splice site selection by promoting splicing of exons with weak 5′ splice sites. Isoform 2 demonstrates enhanced splicing regulatory activity as compared to Isoform 1. Most prominently expressed in heart, small intestine, kidneys, liver, lungs, skeletal muscle, pancreas, ovary, and testis. Disease associated. *	*Nuc, Cyto,* *Stress granule*	***5*** *trscpts/**5** isoforms*	***11*** *(uc)*
*** TRA2A** **(Q13595)**	*TRA2A*	*Ch.7p15.3*	*Associates with pre-mRNA in a sequence-specific manner to regulate pre-mRNA splicing. Expression is ubiquitous. Associates with SR30, SRSF3, SRSF4, SRSF5, and SRSF6.*	*Nuc*	***3*** *trscpts/**3** isoforms*	***63*** *(uc)*
*** TRA2B** **(P62995)**	*TRA2B*	*Ch.3q27.2*	*Associates with pre-mRNA in a sequence-specific manner to promote or inhibit exon inclusion. Works by antagonizing splicing regulators belonging to the hnRNP class of proteins. Ubiquitously expressed with highest expression in heart, skeletal muscle, and pancreas; lowest expression in kidneys and liver.*	*Nuc*	***2*** *trscpts/**2** isoforms*	***67*** *(uc)*

Data and information presented in the table were derived from UniProt/SwissProt (https//:www.uniprot.org accessed on 18 May 2023), PhosphositePlus (https://www.phosphosite.org accessed on 22 May 2023), ENSEMBL (https://www.ensembl.org accessed on 19 May 2023) and Ref. [26]. Expression of all the proteins unless otherwise stated under “Function/Role” is ubiquitous. ^1^ Alternate transcripts (protein coding only) and protein isoforms are defined as those currently verified and annotated in ENSEMBL, UniProt/SwissProt, and NCBI databases. In some cases, (#) verified transcripts have no corresponding verified protein annotated and *vice versa*. ^2^ Post-translational modifications (PTMs) were retrieved from PhosphositePlus (https://www.phosphosite.org accessed on 22 May 2023) and include acetylations, caspase cleavage, *O*-linked β-*N*-acetylglucosaminylation (O-GlcNAc), methylations, phosphorylations, succinylations, sumoylations, and ubiquitinations. The level to which these PTMs have been studied is defined in parentheses next to the number of PTMs identified for each protein: **uc**—uncharacterized and **muc**—mostly uncharacterized. * This protein is not considered part of the SR splicing factor family but is included here for its role in regulating members of this family.

**Table 2 genes-14-01386-t002:** Heterogeneous Nuclear Ribonucleoproteins: Mediators of alternative splicing.

*Protein* (*UniProt ID*)	*Gene*	*Location*	*Function*/*Role*	*Localization*	^ 1 ^ *Alternant* *Transcripts*/*Isoforms*	^ 2 ^ *PTMs*
**FUS** **(P35637)**	*FUS*	*Ch.16p11.2*	*Also referred to as **hnRNP P2**. DNA/RNA-binding protein that plays a role in transcription regulation, RNA splicing, RNA transport, and DNA damage response. Acts as a molecular mediator between RNA polymerase II and U1 small nuclear ribonucleoprotein, thus coupling transcription and splicing. Autoregulates its own expression through nonsense mediated decay. **Disease associated.***	*Nuc*	***2*** *trscpts/**2** isoforms*	***98*** *(**46** wc)*
**ROA0** **(Q13151)**	*HNRNPA0*	*Ch.5q31.2*	*Specifically binds AU-rich element (ARE)-containing mRNAs and is involved in post-transcriptional stability of diverse cytokine mRNAs.*	*Nuc*	***1*** *trscpt/**1** isoform*	***62*** *(muc)*
**ROA1** **(P09651)**	*HNRNPA1*	*Ch.12q13.13*	*Involved in the packaging of pre-mRNA into hnRNP particles, nucleo-cytoplasmic transport of poly(A) mRNA, and modulation of splice site selection. **Disease associated**.*	*Nuc, Cyto* *(shuttles)*	***5*** *trscpts/**3** isoforms*	***91*** *(ssc)*
**RA1L2** **(Q32P51)**	*HNRNPA1L2*	*Ch.13q14.3*	*Involved in the packaging of pre-mRNA into hnRNP particles, nucleo-cytoplasmic transport of poly(A) mRNA, and modulation of splice site selection.*	*Nuc, Cyto*	***1*** *trscpt/**1** isoform*	***37*** *(uc)*
**ROAA** **(Q99729)**	*HNRNPAB*	*Ch.5q35.3*	*Has high affinity for G-rich and U-rich regions of ss hnRNA. Binds to APOB transcripts around the APOBEC C-to-U RNA editing site.*	*Nuc, Cyto*	***4*** *trscpts/**4** isoforms*	***43*** *(uc)*
**ROA2** **(P22626)**	*HNRNPA2B1*	*Ch.7p15.2*	*Associates with pre-mRNAs in a sequence-dependent fashion to package the transcripts. Packaging plays a role in transcription, pre-mRNA processing, RNA nuclear export, subcellular location, mRNA translation, and stability of mature mRNAs. Promotes pri-miRNA processing and sorting. Involved in innate immune response activation. **Disease associated**.*	*Nuc, Cyto,* *Nucleoplasm, Cyto granules, secreted*	***7*** *trscpts/**2** isoforms*	***107*** *(muc)*
**ROA3** **(P51991)**	*HNRNPA3*	*Ch.2q31.2*	*Has a role in cytoplasmic trafficking of RNA. May be involved in pre-mRNA splicing.*	*Nuc*	***5*** *trscpts/**2** isoforms*	***94*** *(muc)*
**HNRPC** **(P07910)**	*HNRNPC*	*Ch.14q11.2*	*Nucleates assembly of 40S hnRNP particles by binding pre-mRNA. May play a role in the early steps of spliceosome assembly and pre-mRNA splicing. N6-methyladenosine (m6A) has been shown to influence mRNA splicing by enhancing HNRNPC binding.*	*Nuc*	***9*** *trscpts/**4** isoforms*	***82*** *(ssc)*
**HNRC1** **(O60812)**	*HNRNPCL1*	*Ch.1p36.21*	*May play a role in nucleosome assembly. Specifically found in skeletal muscle tissue.*	*Nuc*	***1*** *trscpt/**1** isoform*	***23*** *(uc)*
**HNRC2** **(B2RXH8)**	*HNRNPCL2*	*Ch.1p36.21*	*May play a role in nucleosome assembly. Tissue expression is unclear.*	*Nuc*	***1*** *trscpt/**1** isoform*	***4*** *(uc)*
**HNRC3** **(B7ZW38)**	*HNRNPCL3*	*Ch.1p36.21*	*Role unknown. Expressed mainly in the cortical plate, blood, and hindlimb stylopod muscle.*	*Nuc*	***1*** *trscpt/**1** isoform*	***11*** *(uc)*
**HNRPD** **(Q14103)**	*HNRNPD*	*Ch.4q21.22*	*Binds with high affinity to AU-rich elements (AREs) found within the 3′-UTR of many proto-oncogenes and cytokine mRNAs. Also functions as a transcription factor. May be involved in translationally coupled mRNA turnover.*	*Nuc, Cyto (shuttles with circadian clock)*	***5*** *trscpts/**4** isoforms*	***76*** *(muc)*
**HNRDL** **(O14979)**	*HNRNPDL*	*Ch.4q21.22*	*Promotes transcriptional activation or repression in a context-dependent manner. Binds AU-rich elements (AREs) found within the 3′-UTR of many proto-oncogenes and cytokine mRNAs with high affinity. Preferentially expressed in heart, brain, placenta, lungs, liver, skeletal muscle, kidneys, pancreas, spleen, thymus, prostate, testis, ovary, small intestine, colon, and leukocytes. **Disease associated.** Elevated expression observed in diverse cancers.*	*Nuc, Cyto* *(shuttles)*	***6*** *trscpts/**3** isoforms*	***61*** *(uc)*
**HNRPF** **(P52597)**	*HNRNPF*	*Ch.10q11.21*	*Component of the heterogeneous nuclear ribonucleoprotein (hnRNP) complexes. Plays a role in regulating alternative splicing events. Maintains target RNA in an unfolded state.*	*Nuc, Nucleoplasm*	***6*** *trscpts/**1** isoform*	***65*** *(muc)*
**HNRH1** **(P31943)**	*HNRNPH1*	*Ch.5q35.3*	*Component of the heterogeneous nuclear ribonucleoprotein (hnRNP) complexes. Regulates pre-mRNA alternative splicing. **Disease associated**.*	*Nuc, Nucleoplasm*	***4*** *trscpts/**2** isoforms*	***64*** *(muc)*
**HNRH2** **(P55795)**	*HNRNPH2*	*Ch.Xq22.1*	*Component of the heterogeneous nuclear ribonucleoprotein (hnRNP) complexes. **Disease associated**.*	*Nuc, Nucleoplasm*	***1*** *trscpt/**1** isoform*	***48*** *(muc)*
**HNRH3** **(P31942)**	*HNRNPH3*	*Ch.10q21.3*	*Participates in early heat shock-induced splicing arrest. Different isoforms suspected of possessing distinct functions in splicing.*	*Nuc*	***2*** *trscpts/**6** isoforms*	***60*** *(uc)*
**HNRPK** **(P61978)**	*HNRNPK*	*Ch.9q21.32*	*A major pre-mRNA-binding protein that binds to poly(C) sequences. Plays a vital role in the p53/TP53 response to DNA damage. **Disease associated**.*	*Nuc, Cyto, Nucleoplasm, Cellular Projections*	***15*** *trscpts/**3** isoforms*	***132*** *(**23** wc)*
**HNRPL** **(P14866)**	*HNRNPL*	*Ch.19p13.2*	*Component of heterogeneous nuclear ribonucleoprotein (hnRNP) complexes. Binds to exonic or intronic sites to function as an activator or repressor of exon inclusion. Regulates its own expression through the inclusion of a "poison" exon.*	*Nuc, Cyto,* *Nucleoplasm*	***2*** *trscpts/**2** isoforms*	***90*** *(ssc)*
**HNRPM** **(P52272)**	*HNRNPM*	*Ch.19p13.2*	*Pre-mRNA binding protein involved in splicing. Acts as a receptor for carcinoembryonic antigen in Kupffer cells to initiate signaling events, leading to the induction of IL-1 alpha, IL-6, IL-10, and TNFα.*	*Nuc, Nucleolus*	***3*** *trscpts/**2** isoforms*	***125*** *(muc)*
**HNRPR** **(O43390)**	*HNRNPR*	*Ch.1q36.12*	*Component of ribonucleosomes. Role unknown. **Disease associated**.*	*Nuc, Cyto,* *Nucleoplasm, Microsome*	***6*** *trscpts/**5** isoforms*	***76*** *(uc)*
**HNRPU** **(Q00839)**	*HNRNPU*	*Ch.1q44*	*DNA-/RNA-binding protein involved in nuclear chromatin organization, telomere-length regulation, transcription, mRNA alternative splicing and stability, transcriptional silencing, and mitotic cell progression. Required for embryonic development. **Disease associated**.*	*Nuc, Cyto, Nuc Matrix, Speckles, cytoskeleton*	***3*** *trscpts/**3** isoforms*	***185*** *(ssc)*
**PCBP1** **(Q15365)**	*PCBP1*	*Ch.2p13.3*	*Also referred to as **hnRNP E1**. Binds ssRNA and ssDNA. Together with PCBP2, may regulate erythropoiesis through mRNA splicing. Predominantly expressed in skeletal muscle, thymus, and peripheral blood leukocytes; lower expression observed in prostate, spleen, testis, ovary, small intestine, heart, liver, and adrenal and thyroid glands.*	*Nuc, Cyto* *(shuttles)*	***1*** *trscpt/**1** isoform*	***49*** *(muc)*
**PCBP2** **(Q15366)**	*PCBP2*	*Ch.12q13.13*	*Also referred to as **hnRNP E2**. Binds ssRNA and ssDNA. Negatively regulates innate immune antiviral responses mediated by MAVS, and the cGAS-STING pathway. Together with PCBP1, may regulate erythropoiesis through mRNA splicing.*	*Nuc, Cyto* *(shuttles)*	***9*** *trscpts/**8** isoforms*	***42*** *(ssc)*
**PTBP1** **(P26599)**	*PTBP1*	*Ch.19p13.3*	*Polypyrimidine tract binding protein (PTBP)-1 (also known as **hnRNP I**) interacts with polypyrimidine (PP) stretches at the branch point region. Plays a role in pre-mRNA splicing, alternative splicing, and alternate 5′-3′ splice site usage. Promotes exon skipping of its own pre-mRNA during muscle cell differentiation. Can sequester miRNAs.*	*Nuc*	***4*** *trscpts/**4** isoforms*	***65*** *(uc)*
**RBMX** **(P38159)**	*RBMX*	*Ch.Xq26.3*	*Also referred to as **hnRNP G**. Involved in the regulation of pre- and post-transcriptional processes. A component of the supraspliceosome complex that regulates pre-mRNA alternative splice site selection. Can activate or suppress exon inclusion. Ubiquitously expressed. **Disease associated**.*	*Nuc*	***3*** *trscpts/**3** isoforms*	***118*** *(uc)*
**HNRPQ** **(O60506)**	*SYNCRIP*	*Ch.6q14.3*	*Component of the CRD-mediated complex. **Isoform 1** binds to APOB mRNA AU-rich sequences and is a regulatory part of the APOB mRNA editosome complex. May be involved in translationally coupled mRNA turnover. **Isoform 3** is a component of the GAIT (gamma interferon-activated inhibitor of translation) complex, which mediates interferon-gamma-induced transcript-selective translation inhibition in inflammation processes. The GAIT complex binds GAIT elements in the 3′-UTR of diverse inflammatory mRNAs to suppress their translation.*	*Nuc, Cyto,* *Microsome, ER* *(Isoforms 1-3 preferentially localize in the nucleoplasm)*	***9*** *trscpts/**4** isoforms*	***87*** *(uc)*

Data and information presented in the table were derived from UniProt/SwissProt (https//:www.uniprot.org accessed on 17 May 2023), PhosphositePlus (https://www.phosphosite.org accessed on 16 May 2023), ENSEMBL (https://www.ensembl.org accessed on 18 May 2023), and Ref. [37]. Expression of all the proteins unless otherwise stated under Function/Role is ubiquitous. ^1^ Alternate transcripts (protein coding only) and protein isoforms are defined as those currently verified and annotated in ENSEMBL, UniProt/SwissProt, and NCBI databases. In some cases, verified transcripts have no verified protein annotated and vice versa. ^2^ Post-translational modifications (PTMs) were retrieved from PhosphositePlus (https://www.phosphosite.org accessed on 18 May 2023) and include acetylations, caspase cleavage (activation/protein processing), methylations, N-glycosylations, *O*-linked β-*N*-acetylglucosaminylation (O-GlcNAc), phosphorylations, succinylations, sumoylations, and ubiquitinations. The level to which these PTMs have been studied is defined in parentheses next to the number of PTMs identified for each protein: **uc**—uncharacterized; **muc**—mostly uncharacterized; **ssc**—some sites modestly characterized; **wc**—a significant number of sites well characterized.

**Table 3 genes-14-01386-t003:** Adenosine and Cytidine Deaminases: RNA editors.

*Protein* (*UniProt ID*)	*Gene*	*Location*	^ 1 ^ *Substrate*	*Function*/*Role*	*Localization*	^ 2 ^ *Alternant* *Transcripts*/*Isoforms*	^ 3 ^ *PTMs*

**ADAR Family of Adenosine Deaminases**
**DSRAD** **(P55265)**	*ADAR*	*Ch.1q21.3*	*dsRNA, Z-DNA*	*Catalyzes the deamination of A-to-I in RNA of complex secondary structure. May affect gene expression by altering protein coding, pre-mRNA splicing, RNA stability, RNA transport, miRNA targeting, and RNA–protein interactions. Edits both host and foreign RNAs. May have a role in DNA strand break repair. **Isoform 1** is interferon-inducible, primarily localizes to the cytoplasm, and demonstrates elevated activity toward foreign (viral) RNAs. Suppresses PKR activation and the induction of the integrated stress response. Can associate with Z-DNA or Z-RNA. **Isoform 5** is constitutively expressed and localizes primarily to the nucleus. Ubiquitously expressed with highest expression in brain and lungs. **Disease associated**. Overexpressed in many cancers; expression associated with cancer aggressiveness.*	*Nuc, Nucleolus, Cyto (shuttles)*	***12*** *trscpts/**5** isoforms #* *These numbers may double due to alternate exon 1 usage.*	***125*** *(ssc)*
**RED1** **(P78563)**	*ADARB1*	*Ch.22q22.3*	*dsRNA*	*Catalyzes the deamination of A-to-I in RNA of complex secondary structure. May affect gene expression by altering protein coding, pre-mRNA splicing, RNA stability, RNA transport, miRNA targeting, and RNA–protein interactions. Edits both host and foreign RNAs. Suppresses PKR activation and the induction of the integrated stress response. Involved in the RNA editing of RNA/DNA hybrids during DNA double-strand break repair. **Disease associated.** Ubiquitously expressed. Highly expressed in brain, heart, lower placenta; moderately expressed in lungs, liver, and kidneys. **Isoform 5** is seen in hippocampus and colon. Decreased activity in astrocytomas correlated with disease severity.*	*Nuc*	***5*** *trscpts/**6** isoforms #*	***23*** *(ssc)*
**RED2** **(Q9NS39)**	*ADARB2*	*Ch.10q15.3*	*dsRNA*	*ADAR family member that lacks editing activity. Likely alters/regulates ADAR1 and ADAR2. Binds both ss and dsRNA. Expression is brain-specific.*	*Nuc*	***2*** *trscpts/**2** isoforms*	***13*** *(uc)*
**Cytidine Deaminases: APOBEC Family and AID**
**ABEC1** **(P41238)**	*APOBEC1*	*Ch.12p13.31*	*ssRNA, ssDNA (h)*	*Catalyzes the C-to-U post-transcriptional editing of several mRNAs, including APOB and NF1. Acts as a homodimer. May play a role in the epigenetic regulation of gene expression. Expressed exclusively in small intestine.*	*Nuc, Cyto*	***1*** *trscpt/**1** isoform #*	***2*** *(wc)*
**ABEC2** **(Q9Y235)**	*APOBEC2*	*Ch.6p21.1*	*ss/dsDNA (h) (binding only)*	*A potential C-to-U editing enzyme with no known substrate and poor deaminase activity. May have a role in epigenetic regulation of gene expression. Exclusively expressed in heart and skeletal muscle.*	*Nuc, Cyto*	***1*** *trscpt/**1** isoform*	***2*** *(uc)*
**ABC3A** **(P31941)**	*APOBEC3A*	*Ch.22q13.1*	*ssRNA, ssDNA*	*Demonstrates C-to-U editing activity toward ssRNA and ssDNA. Best effectiveness against ssDNAs of foreign origin (viral). Deaminates both C and meC and can hyper-edit nuclear and mitochondrial DNA. May have a role in epigenetic regulation of gene expression. Expressed in peripheral leukocytes and CD14^+^ phagocytic cells. Highly expressed in keratinocytes and peripheral blood monocytes. Present at detectable levels in other lymphoid tissue, lungs, bladder, urinary tract, and adipose tissue. Enhanced expression observed in diverse tumor tissues. **Induced by interferon and CpG ssDNA**.*	*Nuc, Cyto*	***3*** *trscpts/**2** isoforms #*	***4*** *(uc)*
**ABC3B** **(Q9UH17)**	*APOBEC3B*	*Ch.22q13.1*	*ssDNA*	*Demonstrates C-to-U editing activity. Selectively targets ssDNAs of foreign origin (viral). Has the ability to hyper-edit nuclear and mitochondrial DNA. Acts as a homodimer. Interacts with APOBEC3G. Expressed in peripheral blood leukocytes, bone marrow, spleen, kidneys, bladder, urinary tract, testes, prostate, heart, thymus, ovary, and gastrointestinal tract. **Expression is induced by interferon**.*	*Nuc*	***3*** *trscpts/**3** isoforms*	***10*** *(muc)*
**ABC3C** **(Q9NRW3)**	*APOBEC3C*	*Ch.22q13.1*	*ssDNA (v)*	*Demonstrates C-to-U editing activity. Selectively targets ssDNAs of foreign origin (viral). May play a role in epigenetic regulation of gene expression. Expressed in peripheral blood mononuclear cells, bone marrow, spleen, thymus, respiratory tract, kidneys, bladder, urinary tract, testes, prostate, skin, muscle, heart, ovary, endocrine tissues, liver, and gastrointestinal tract. **Expression is induced by interferon**.*	*Nuc, Cyto*	***1*** *trscpt/**1** isoform*	***5*** *(uc)*
**ABC3D** **(Q96AK3)**	*APOBEC3D*	*Ch.22q13.1*	*ssDNA (v)*	*Demonstrates C-to-U editing activity. Selectively targets ssDNAs of foreign origin (viral). Antiviral activity occurs through both deaminase-dependent and -independent mechanisms. Can homodimerize or form heterodimers with APOBEC3F and APOBEC3G. Expressed in peripheral blood mononuclear cells, bone marrow, lymphoid tissues, gastrointestinal tract, and female reproductive system.*	*Cyto, P-bodies*	***2*** *trscpts/**2** isoforms*	***4*** *(uc)*
**ABC3F** **(Q8UX4)**	*APOBEC3F*	*Ch.22q13.1*	*ssDNA (v)*	*Demonstrates C-to-U editing activity. Selectively targets ssDNAs of foreign origin (viral). Exhibits antiviral activity toward wide spectrum of viruses through both deaminase-dependent and -independent mechanisms. Has not been shown to target nuclear or mitochondrial DNA. May play a role in epigenetic regulation of gene expression. Widely expressed with highest expression observed in ovary. Interacts with APOBEC3G. **Expression is induced by interferon**.*	*Cyto, P-bodies*	***2*** *trscpts/**3** isoforms #*	***3*** *(muc)*
**ABC3G** **(Q9HC16)**	*APOBEC3G*	*Ch.22q13.1*	*ssRNA, ssDNA (v)*	*Demonstrates C-to-U editing activity. Selectively targets ssDNAs of foreign origin (viral). Antiviral activity occurs through both deaminase-dependent and -independent mechanisms. Acts as a homodimer or homo-oligomer. Can bind RNA and form inactive high molecular weight (HMM) or active low molecular weight (LMM) ribonucleoprotein complexes. Expressed in peripheral blood mononuclear cells, bone marrow, lymphoid tissue, ovary, testis, bladder, urinary tract, gastrointestinal tract, and liver respiratory tract. Interacts with APOBEC3B, APOBEC3F, MOV10, AGO2, EIF4E, EIF4ENIF1, DCP2, and DDX6 in an RNA-dependent manner. **Expression is induced by interferon.** Upregulated in certain tumors.*	*Nuc, Cyto,* *P-bodies*	***1*** *trscpt/**2** isoforms #*	***11*** *(ssc)*
**ABC3H** **(Q6NTF7)**	*APOBEC3H*	*Ch.22q13.1*	*ssDNA (v)*	*Demonstrates C-to-U editing activity. Selectively targets ssDNAs of foreign origin (viral). Antiviral activity occurs through both deaminase-dependent and -independent mechanisms. Expressed in peripheral blood mononuclear cells, bone marrow, lymphoid tissue, lungs, testis, ovary, and skin.*	*Nuc, Cyto,* *P-bodies*	***6*** *trscpts/**4** isoforms*	***1*** *(uc)*
**ABEC4** **(Q8WW27)**	*APOBEC4*	*Ch.1q25.3*	*ND*	*Possible C-to-U editing enzyme with no known substrate. Expression is restricted to testis.*	*Nuc (predicted)*	***1*** *trscpt/**1** isoform*	***1*** *(uc)*
**AICDA** **(Q9GZX7)**	*AICDA*	*12p13.31*	*ssRNA (binding only),* *ssDNA (h),* *RNA/DNA hybrid*	*Demonstrates C-to-U editing activity. Involved in somatic hyper-editing, gene conversion, and class-switch recombination in B-lymphocytes. Necessary for B-cell differentiation and antibody maturation. Hyper-edits other sites in the genome in several pathologies. May have a role in epigenetic regulation of gene expression. **Disease associated**. Highly expressed in lymphoid tissues and germinal center B-cells.*	*Nuc, Cyto (predominant)*	***4*** *trscpts/**4** isoforms*	***7*** *(2 wc)*

Data and information presented in the table were derived from UniProt/SwissProt (https//:www.uniprot.org accessed on 23 May 2023), PhosphositePlus (https://www.phosphosite.org accessed on 25 May 2023), ENSEMBL (https://www.ensembl.org accessed on 24 May 2023), and Refs. [12,42]. Expression of all the proteins unless otherwise stated under “Function/Role” is ubiquitous. ^1^ For substrate nucleic acid, (h)=human; (v)=viral. ^2^ Alternate transcripts (protein coding only) and protein isoforms are defined as those currently verified and annotated in ENSEMBL, UniProt/SwissProt, and NCBI databases. In some cases, (#) verified transcripts have no corresponding verified protein annotated and vice versa. ^3^ Post-translational modifications (PTMs) were retrieved from PhosphositePlus (https://www.phosphosite.org accessed on 18 May 2023) and include acetylations, caspase cleavage (alteration of activity), methylations, phosphorylations, succinylations, sumoylations, and ubiquitinations. The level to which these PTMs have been studied is defined in parentheses next to the number of PTMs identified for each protein: **uc**—uncharacterized; **muc**—mostly uncharacterized; **ssc**—some sites characterized; **wc**—sites well characterized.

## Data Availability

Not applicable.

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
