# Peer review of "Alternative Splicing, RNA Editing, and the Current Limits of Next Generation Sequencing"

_genes, 2023, doi:10.3390/genes14071386_

Round 1

Reviewer 1 Report

The paper by Piazzi and collaborators discuss mechanisms of RNA editing and sought to discuss limitations imposed by this editing in current gene expression analysis platforms.

Overall the theme is of interest. However, much focus is given to the molecular mechanisms of different forms of RNA editing, while its impact on RNA-seq analysis is poorly discussed, being obscured by thorough discussion on the mechanisms. This makes the text hard to follow and disconnected from its title and the aims proposed on the abstract and introduction. When I've read these sections I expected to see more discussion about new methods, platforms and bioinformatic pipelines to cope with the issues presented by mRNA editing throughout the text, which is not done. Discussion about these themes is restricted to small vague phrases through the text and in the conclusion section.

Therefore, in my opinion it would be better to change the focus of the manuscript to discuss only the molecular mechanisms of different forms of RNA editing and maybe discuss their impact of RNA editing in transcriptome analyses in a separate section or in the introduction, but not as a main focus of the manuscript.

Also, tables present long stretches of text, which makes then poorly intuitive and objective. As a suggestion, the text in "Function/Roles" columns may be substituted by topics, key words or more concise texts.

The text may benefit from an English language review, since some phrases are confuse and some tipos are present. 

e.g.: 

Line 210: "demonatrate"

Line 277: "amine base" seems confuse. "Deamination at the C6" position would be sufficient.

Lines 267-270: "Thus, the potential exists for hnRNPs, either through differential expression, isoforms expressed, PTM regulation or mutation, to produce an infinite number potential transcripts and protein isoforms that have specific roles but have not yet been identified or annotated." confuse construction.

Line 393-396: "Unlike ADAR1 and ADAR2, where the APOBEC proteins lack in transcriptional diversity (only 3A, 3B, 394 3D, 3F, 3G, 3H and AID are known to express alternatively spliced isoforms) the APOBEC proteins make-up for in number (Figure 3B and Table 3)." The phrase is confuse and seems to be wrong. 

Author Response

We would like to thank the Reviewer for their time and suggestions. 

Comment:

The paper by Piazzi and collaborators discuss mechanisms of RNA editing and sought to discuss limitations imposed by this editing in current gene expression analysis platforms.

Overall the theme is of interest. However, much focus is given to the molecular mechanisms of different forms of RNA editing, while its impact on RNA-seq analysis is poorly discussed, being obscured by thorough discussion on the mechanisms. This makes the text hard to follow and disconnected from its title and the aims proposed on the abstract and introduction. When I've read these sections I expected to see more discussion about new methods, platforms and bioinformatic pipelines to cope with the issues presented by mRNA editing throughout the text, which is not done. Discussion about these themes is restricted to small vague phrases through the text and in the conclusion section.

Therefore, in my opinion it would be better to change the focus of the manuscript to discuss only the molecular mechanisms of different forms of RNA editing and maybe discuss their impact of RNA editing in transcriptome analyses in a separate section or in the introduction, but not as a main focus of the manuscript.

Response:

We agree with the reviewer on this observation, thus we have attempted to remedy this by adding two new sections 3.3 and 4.3 which discuss, respectively, the consequences that splicing factors (SRSFs and hnRNPs) and RNA editing can have on RNA-seq transcriptome analysis and the current remediating protocols/pipelines in practice and development to deal with the issues raised.

Comment:

Also, tables present long stretches of text, which makes then poorly intuitive and objective. As a suggestion, the text in "Function/Roles" columns may be substituted by topics, key words or more concise texts.

Response:

Our objective with the tables "Function/Roles" column was to serve as a reference, presenting important information not present in the text, such as tissue expression, direct association with disease, highlight the differing functions between isoforms (if known), enzymatic function (as a monomer, dimer, tetramer), important protein-protein interactions (if known) and highlight the association that several of the discussed proteins have with the innate immune response. But we agree that the wording in some cases was verbose or presented slightly extraneous information as regards this manuscript, so the tables were edited to shorten the text under "Function/Roles".

We also are requesting that the editors print these tables in any final version horizontally to enhance the legibility. Unfortunately, adding these in the original submission was not possible, and instead, horizontal versions of the tables were added as supplemental data.   

Comment:

Comments on the Quality of English Language

The text may benefit from an English language review, since some phrases are confuse and some tipos are present. 

e.g.: 

Line 210: "demonatrate"

Line 277: "amine base" seems confuse. "Deamination at the C6" position would be sufficient.

Lines 267-270: "Thus, the potential exists for hnRNPs, either through differential expression, isoforms expressed, PTM regulation or mutation, to produce an infinite number potential transcripts and protein isoforms that have specific roles but have not yet been identified or annotated." confuse construction.

Line 393-396: "Unlike ADAR1 and ADAR2, where the APOBEC proteins lack in transcriptional diversity (only 3A, 3B, 394 3D, 3F, 3G, 3H and AID are known to express alternatively spliced isoforms) the APOBEC proteins make-up for in number (Figure 3B and Table 3)." The phrase is confuse and seems to be wrong. 

Response:

We apologize for these oversights on our part. We have addressed each of the issues raised by the Reviewer and have thoroughly controlled the text and tables for typos and proofread the revised manuscript.

We sincerely thank the Reviewer for their time and attention to this manuscript and their helpful suggestions.

Kindest regards,

William Blalock, PhD

Research Scientist

IGM-CNR/Rizzoli Orthopedic Institute

via di Barbiano 1/10

40136 Bologna, Italy

Reviewer 2 Report

     This comprehensive review (that is called by the authors, with a touch of false modesty, "brief perspective" /line 23/) describes the mechanisms of alternative splicing and RNA editing. The main focus of the manuscript is on the proteins involved in these posttranscriptional processes: Ser/Arg-rich splicing factors, heterogenous nuclear RNPs, adenosine deaminases, cytidine deaminases. Characteristic features of these proteins (expression patterns, cellular localisation,, their regulation by various posttranslational modifications are described in the text and summarized in Tables /that, unfortunately, are quite hard to read/). The review presents an extremely complex picture in which the many white spots are also emphasized. The authors also point out that the existing oligonucleotide-array and NGS transcriptomic databases should be continuously updated.

     The text contains typos and other errors that need to be corrected:

line 136:  hnRNP stands for heterogenous nuclear ribonucleoprotein, the term should be used consistently.

line 154:  ...and regulators...

line 176:  ...main SRSF (A) and hnRNP (B) proteins involved...

line 210:  ...demonstrate...

line 218:  ...infinite number of...

line 257:  ...hnRNPs...

line 285:  ...cytidine deaminases.

line 295:  ...of family members... - should be deleted.

line 383:  ...panorama...

line 412:  ...apolipoprotein B...

line 459:  confusing sentence, should be corrected.

Author Response

We would like to thank the Reviewer for their time and suggestions. 

Comment:

This comprehensive review (that is called by the authors, with a touch of false modesty, "brief perspective" /line 23/) describes the mechanisms of alternative splicing and RNA editing. The main focus of the manuscript is on the proteins involved in these posttranscriptional processes: Ser/Arg-rich splicing factors, heterogenous nuclear RNPs, adenosine deaminases, cytidine deaminases. Characteristic features of these proteins (expression patterns, cellular localisation,, their regulation by various posttranslational modifications are described in the text and summarized in Tables /that, unfortunately, are quite hard to read/). The review presents an extremely complex picture in which the many white spots are also emphasized. The authors also point out that the existing oligonucleotide-array and NGS transcriptomic databases should be continuously updated.

Response:

We thank the Reviewer for their kind comments. As regards the tables, we agree and are sorry for the poor legibility. Adding the tables horizontally was not possible during the submission process; we are asking the editors to print them horizontally if possible in any published version. To compensate, we have added the tables, formatted horizontally with larger font print as supplemental data.     

Comment:

The text contains typos and other errors that need to be corrected:

line 136:  hnRNP stands for heterogenous nuclear ribonucleoprotein, the term should be used consistently.

line 154:  ...and regulators...

line 176:  ...main SRSF (A) and hnRNP (B) proteins involved...

line 210:  ...demonstrate...

line 218:  ...infinite number of...

line 257:  ...hnRNPs...

line 285:  ...cytidine deaminases.

line 295:  ...of family members... - should be deleted.

line 383:  ...panorama...

line 412:  ...apolipoprotein B...

line 459:  confusing sentence, should be corrected.

Response:

We sincerely thank the Reviewer for their careful attention to the text. We have made all the suggested corrections and have thoroughly checked for typos and proofread the revised version.

Again we would like to thank the Reviewer for their time and helpful suggestions.

Kindest regards,

William Blalock, PhD

Research Scientist

IGM-CNR/Rizzoli Orthopedic Institute

via di Barbiano 1/10

40136 Bologna, Italy
